# A UV-Protective Textile Coating Based on Recycled Poly(vinyl butyral) (PVB): A New Life for a Waste Polymer

**DOI:** 10.3390/polym16233439

**Published:** 2024-12-07

**Authors:** Noemi Cei, Ilaria Canesi, Stefano Nejrotti, Giorgia Montalbano, Hamideh Darjazi, Alessandro Piovano, Matteo Bonomo, Alberto Fina, Beatriz Yecora, Angelica Perez, Claudia Barolo, Claudio Gerbaldi, Daniele Spinelli

**Affiliations:** 1Next Technology Tecnotessile, Via del Gelso 13, 59100 Prato, Italy; noemi.cei@tecnotex.it (N.C.); daniele.spinelli@tecnotex.it (D.S.); 2Department of Chemistry, NIS Interdepartmental Center and INSTM Ref. Centre, University of Torino, 10125 Turin, Italy; stefano.nejrotti@unito.it (S.N.); matteo.bonomo@unito.it (M.B.); claudia.barolo@unito.it (C.B.); 3Istituto di Scienza, Tecnologia e Sostenibilità per lo Sviluppo dei Materiali Ceramici (ISSMC-CNR), Via Granarolo 64, 48018 Faenza, Italy; 4Department of Applied Science and Technology (DISAT), Politecnico di Torino, Corso Duca Degli Abruzzi 24, 10129 Turin, Italy; giorgia.montalbano@polito.it (G.M.); hamideh.darjazi@polito.it (H.D.); claudio.gerbaldi@polito.it (C.G.); 5National Reference Centre for Electrochemical Energy Storage (GISEL)–INSTM, Via G. Giusti 9, 50121 Florence, Italy; 6Department of Applied Science and Technology (DISAT), Politecnico di Torino, Viale Teresa Michel 5, 15121 Alessandria, Italy; alberto.fina@polito.it; 7LUREDERRA Technological Centre, Perguita Industrial Area, 31210 Los Arcos, Spain; beatriz.yecora@lurederra.es (B.Y.); angelica.perez@lurederra.es (A.P.)

**Keywords:** coating, textile substrate, secondary raw material, end of life, sustainability

## Abstract

Polyvinyl butyral (PVB) is a high-performance thermoplastic polymer, commonly used as an interlayer material in laminated safety glass for the automotive and architectural sectors. Currently, there is no end-of-life cycle program for a substantial amount of PVB film, which mainly ends up in landfills. According to a circular approach, PVB can be revalorized after efficient separation and recovery from glass. Thus, the aim of this work was to develop functional coatings for textile applications using recycled PVB (re-PVB), also in combination with an organic UV absorber, to enable the production of UV-protective final coated fabrics. The re-PVB-coated fabrics were obtained through an industrially scalable spraying process (leading to an average weight increase of 20 ± 3 wt.%), and the effectiveness of the application was evaluated according to different characterization techniques, such as FT-IR (Fourier transform–infrared) spectroscopy, SEM (scanning electron microscope), a washing test, a mechanical test, a thermo-physiological test, and the ultraviolet protection factor (UPF). Based on the results, the re-PVB-coated fabrics appeared stable upon washing (with a negligible weight loss compared to the average amount of coating) and effective in UV protection (with a final UPF being four times higher and a reduced UVA transmittance from 2.0% to 0.6%).

## 1. Introduction

The textile and clothing industry is among the largest industrial branches in the EU. In 2023, the overall size of the textile and clothing industry in the EU was represented by a turnover of EUR 170 billion and investments of EUR 4.5 billion [1]. In addition, the European production of technical textiles—such as UV-protective textiles—is currently very important, with a growing share of 29% in textile production. However, the consumption of textiles in the EU has the fourth-highest negative impact on the environment and on climate change. According to a communication from the European Commission, the global consumption of clothing and footwear is expected to increase by 63% by 2030 [2]. A report by McKinsey [3] highlights that, in 2020, the textile field was among the top five sectors with the highest plastic consumption, accounting for around 90% of the global volume of plastic demand. In this scenario, taking into account possible sustainable solutions that are in line with the principles of the circular economy, the recycling of plastic waste has the potential to assume a pivotal role in the textile field [4].

In the present day, thermoplastic polymer-based waste is usually recycled for the realization of novel fibers, but it can also have a positive impact on the development of more sustainable coatings on textile products. Besides aesthetic reasons, textiles are often coated to enhance some properties of materials, such as flame retardancy, UV protection, easy-care surfaces, water and oil repellence, and antimicrobial properties [5,6]. A typical coating formulation contains polymeric binder(s) along with other additives, which can be applied to the textile through different coating techniques (e.g., spray coating, pad coating, and knife coating). The properties of the final product depend on the type of polymer used and its formulation, as well as the nature of the substrate and the coating method. However, one of the issues of the textile sector is related to traditional textile finishing, for which there is a massive use of potentially harmful materials, such as in treatments with polyvinyl chloride (PVC) or with polyurethane (PU) [7,8,9]. Different environmental and safety legislations have been introduced by various governments, including regulations from the European Commission such as REACH, to ensure the safe use of coated products throughout their entire life cycle [10]. Therefore, researchers and companies are increasingly focusing on the development of new products to obtain sustainable alternatives to traditional polymer-based textile coatings [11].

An alternative polymer for this application could be poly (vinyl butyral) (PVB). PVB is a high-performance thermoplastic polymer, commonly used as the interlayer in laminated safety glass for automotive and architectural purposes (Appendix A). It demonstrates excellent mechanical strength, high-temperature resistance, strong adhesion to various substrates, UV stability, and optical transparency [12,13,14].

PVB is a vinyl acetal polymer, and its synthesis starts with the partial hydrolysis of poly (vinyl acetate) (PVAc) to poly (vinyl alcohol) (PVA). The final step consists of a condensation reaction between the hydroxyl groups of PVA and butyraldehyde (BA) in the presence of an acid catalyst. The resulting polymer presents a random structure with segments made of vinyl butyral units (VB, 76–80 wt.%), vinyl alcohol units (VA, 18–24 wt.%), and vinyl acetate units (VAc, 1–4 wt.%) [12,13,14,15]. The final properties of PVB depend on many factors connected to the production process and to the processability conditions, which determine the exact structure of the polymer in terms of the length of the chains, percentage, and order of the different units. The content of VA units mainly affects the adhesion properties, the crosslinking behavior, and the miscibility of blends, while the number of VB units usually influences the hydrophilicity and the mechanical and thermal properties of PVB [16,17]. PVB is soluble in many organic solvents, but it can be crosslinked with various chemicals in order to improve its solvent, chemical, and thermal resistance. Normally, the crosslinking capacity of PVB depends on the number of available hydroxyl groups that can undergo condensation reactions with phenolic, melamine, and epoxy resins, along with isocyanates [13,14,16].

Since unplasticized vinyl acetal polymers are brittle and unpliable, making them more difficult to process, PVB for lamination is generally prepared as a film by melt mixing with plasticizers (15–30 wt.%) [14]. The typology and the content of plasticizers are essential, especially concerning their mechanical and rheological properties. The plasticizer must be compatible and soluble with the polymer, and this miscibility is greatly influenced by the VA content of the polymer. Phthalates have been used, but regulations have restricted the use of these additives for health and environmental reasons [18,19]. Phthalate-free plasticizers are currently added to the PVB matrix, mostly esters and ethers compounds, some of which are triethyleneglycol di-(2-ethyl hexanoate) or tetraethylene glycol di-*n*-heptanoate, dibutyl sebacate, dihexyl adipate, dioctyl adipate, hexyl cyclohexyl adipate, or mixtures of heptyl and nonyl adipates [12].

The global PVB market is projected to grow considerably in the coming years due to increasing industrialization and urbanization, with a market size reaching USD 3641.25 million by 2031 [20]. Worldwide, 65% of all PVB films are used for vehicle windshield applications, considering that the total amount of PVB film produced for the automotive and construction industries is equal to 170 million kg per year [21]. However, there is no end-of-life cycle program for a substantial amount of PVB film, which mainly ends up in landfills. Indeed, the main issues for the recycling of laminated PVB are the removal of glass from the polymer, the partial degradation of PVB due to UV radiation and heat, and the unavoidable reprocessing step. Therefore, the need to recycle a valuable waste stream such as PVB could gain more importance, according to a circular and sustainable perspective. The use of re-PVB not only has environmental benefits, with an estimated carbon footprint 25 times lower than the virgin material, but could also be used in new applications to obtain various properties of the polymer [16,22].

For instance, thanks to its adhesion properties, PVB can be repurposed as a binder or as a matrix in combination with filling materials for coating production. Grethe et al. [13] successfully integrated organic or inorganic UV absorbers into the PVB matrix to realize effective UV-protective coatings on textiles. The absorption characteristics of the developed PVB coatings depend on different factors, such as the type, the distribution into the matrix, and the intrinsic absorption of the additive, offering the possibility of realizing tailored UV-protective coatings. Another potential PVB application in the textile field could be as backing for carpets and floor coverings, where the function of PVB is to secure the pile yarns to structures [8]. PVB is also applied in the development of smart and e-textiles. Roshni et al. [23] reported an example of a wearable PVB-coated textile antenna, obtaining a thin, flexible, and water-resistant product that can be potentially integrated into new-generation clothing. In addition, examples of blends of PVB with inorganic nanoparticles to produce antibacterial coatings on textiles can be found in the literature [8,24,25]. Although interest in PVB recovery and recycling is growing, the above-described applications mainly used virgin material or blends containing a small quantity of re-PVB. Conversely, in this report, functional coatings on textiles are developed using re-PVB only.

Since clothing is the most convenient way of protecting the human body from external environmental factors, UV-protective clothes represent a significant part of the technical textile sector. Even if the most dangerous UV-C radiation (100–290 nm) is absorbed by the ozone layer, prolonged exposure to UV-A (315–400 nm) and UV-B (290–315 nm) radiation can cause several skin problems, such as skin aging, skin reddening, erythema, and skin cancer [26,27]. The possibilities of textile materials for UV radiation protection are being illustrated by an increasing number of studies [28,29,30,31]. Any fabric should either absorb or reflect the UV radiation, thus limiting its transmission to the human body. According to the standard test method UNI EN 13758-1:2007, the UPF provides a measure of the efficiency of a textile material to protect the skin from excessive harmful UV radiation. A high value of the UPF means that the amount of UV radiation transmitted through the textile is low: UPF values above 50 are considered as excellent safe “sun-blockers”. Several studies have revealed that the main factors influencing the ability of blocking harmful UV radiation include the fiber type, fabric structure (e.g., thickness, density, and porosity), type of dye, and finishing treatments (e.g., the use of organic or inorganic UV absorbers) [26,27,29,30]. Thus, from a favorable combination of these factors, it is possible to design and develop fabrics with improved UV protection performances.

Considering the outstanding properties of PVB films—such as their good light resistance, excellent transparency, and excellent adhesive properties [16]—and the limited examples in the literature connected to the use of the recycled polymer, in this work, re-PVB was used as a polymer matrix to develop functional coatings for textiles. The addition of an organic UV absorber into the re-PVB formulation could enhance the UV protection performance of the final coated textile. A synthetic fabric was chosen as a textile substrate since 60% of the global market of textiles is covered by synthetic fibers (55% by polyester and 5% by polyamides), followed by cotton (22%) [32]. The developed re-PVB formulations were firstly analyzed in terms of visco-elastic properties and applied to the fabric through an efficient and easily scalable process, i.e., the spray-coating technique. The re-PVB-coated fabrics were then characterized using FT-IR and SEM analysis, as well as washing, mechanical, and thermo-physiological tests. To assess the effectiveness of the material as a UV protector, the UPF value of the re-PVB-coated fabrics was also measured.

## 2. Materials and Methods

Plain woven synthetic textile (87% polyester + 13% elastane) was selected and purchased from Oceano Tex (Prato, Italy). The selected fabric had a thickness of 0.42 mm and a surface mass of 181 g/m^2^. The re-PVB was obtained through a special mechanical and chemical process developed by Lurederra Technological Centre [12,33]. Nikitakos et al. [12] have thoroughly characterized the same re-PVB supplied by Lurederra Technological Centre. Ethanol absolute (analytical grade ≥ 99%) used as a solvent was purchased from Prokeme srl (Prato, Italy). The blocked isocyanate crosslinking agent RucoCoat FX 8041 was kindly provided by Nextore srl (Prato, Italy). According to the TDS, this product complies with different textile certifications and programs (i.e., Bluesign^®^, Standard 100 by OEKO-TEX^®^, and ZDHC Gateway—Conformity level 3). The organic UV absorber SemaSORB^®^ UV20107 was supplied by SEMA Gmbh (Coswig, Germany). The chemical structure of SemaSORB^®^ UV20107 is shown in Figure 1. All commercial chemicals were used without any additional purification in this study. Deuterated methanol (CD_3_OD, >99.8 atom % D) was purchased from Merck (Darmstadt, Germany).

### 2.1. Preparation of re-PVB Formulations for Coating Application

The re-PVB was dissolved into ethanol under stirring and heating conditions, achieving a concentration of 5 wt.%. This re-PVB formulation was identified as re-PVB_0. The temperature reached for a better re-PVB dissolution was approximately equal to 50 °C. Considering the results obtained by Brendgen, R. et al. [8], the final re-PVB formulation for the coating application was obtained by adding the crosslinking agent (i.e., RucoCoat FX 8041) to the previous formulation at a concentration of 10 wt.%. This was the lowest tested amount of cross-linking agent, which represents the best compromise between performance and cost. The final re-PVB formulation was identified as re-PVB_1. Considering research in the literature [13,28] and the products available on the market, the organic UV absorber SemaSORB UV20107 was also tested in the re-PVB_1 formulation at a concentration of either 0.5 wt.% or 1 wt.%. These re-PVB formulations were identified as re-PVB_2 and re-PVB_3, respectively.

### 2.2. Rheology of re-PVB Formulations

Rheological tests were performed using a DHR-2 controlled stress rotational Rheometer (TA Instruments−Waters LLC, New Castle, DE, USA) equipped with a parallel plate geometry with a diameter of 20 mm and a Peltier plate system to constantly control the system temperature. For each experiment, the initial gap was adjusted according to the sample thickness (ranging from 800 and 1200 μm). The viscosity of the re-PVB formulations (i.e., re-PVB_0, re-PVB_1, and re-PVB_3) was measured by performing flow ramp tests at increasing shear rates in the range of 0.01 to 500 s^−1^ at a constant temperature of 25 °C.

Temperature ramp tests between 20 °C and 40 °C with a rate of 3 °C/min were also carried out to explore any potential dependence of the visco-elastic properties of the solutions on temperature, setting an oscillation stress at a 20% amplitude and 1 Hz frequency.

The mean curve and standard deviation were obtained by measuring three samples of each formulation for each experiment.

### 2.3. Coating Application Process

To verify the effectiveness of the crosslinking reaction, a re-PVB_1 film was prepared by casting the formulation on a flat substrate and then by heating it at 160 °C for 6 min in an oven. The same procedure was followed for re-PVB_2 film and re-PVB_3 film.

Spray coating is considered a common direct-coating technique in the textile sector [5,6]; a spray gun is used to nebulize and deposit a solution/dispersion directly onto a textile substrate. This technique permits the better control of the quantity of the solution/dispersion deposited on the surface and the treatment of only one side or both sides of the material. The two-step spraying treatment developed in this work (Figure 2) could be considered an easily scalable industrial process, which allows the use of re-PVB for an alternative application. The coating application was carried out by spraying the re-PVB formulations (i.e., re-PVB_1, re-PVB_2, and re-PVB_3) on the polyester samples using a spray gun (nozzle diameter = 1.5 mm). The formulations were pre-heated to ensure effective homogenization, and the application temperature was about 40 °C. The samples were placed on a horizontal holder, maintaining the spray gun at a constant distance of 6–8 cm. After the coating application, the samples were dried in the oven at 160 °C for 6 min. The mean weight increase for all the coated samples was 20 ± 3 wt.%, registering an average surface mass of 218 g/m^2^. Three types of coated samples were produced: re-PVB_1-coated fabric, re-PVB_2-coated fabric, and re-PVB_3-coated fabric.

### 2.4. Evaluation of Coated Textile Properties

#### Fourier Transform–Infrared (FT-IR) Spectroscopy

Attenuated total reflection (ATR) FT-IR spectra were registered using a SHIMADZU FTIR IRAffinity-1S spectrometer (Shimadzu Corporation, Kyoto, Japan) from 400 to 4000 cm^−1^ with a 4 cm^−1^ spectral resolution, equipped with a diamond crystal. For each uncoated and coated textile sample, many FT-IR spectra in different spots were acquired so that the reported spectra could be considered statistically significant.

### 2.5. Microscopic Analysis

SEM (scanning electron microscope) images of the samples were obtained using an FEI Quanta 200 FEG (low-vacuum mode, 5–10 kV acceleration voltage) and an ESEM Quanta 400 (FEI) (high-vacuum mode, 20 kV acceleration voltage) (Thermo Fisher Scientific, Waltham, MA, USA). Before the analysis, samples were placed on AI SEM stubs by the means of carbon adhesive disks and sputter coated with a Au/Pd alloy. Optical microscope images of the samples were obtained using a Nikon Leica MS 5 (Leica Microsystems, Wetzlar, Germany), with a magnification of 10×.

### 2.6. Nuclear Magnetic Resonance (NMR)

NMR spectra were acquired using a Jeol ECZR600 spectrometer, working at 14.1 T (^1^H operating frequency: 600 MHz), in CD_3_OD, using Wilmad high-throughput Class B Glass NMR tubes. The residual solvent peak was used as an internal reference (CH_3_OH, ^1^H: 3.31 ppm). The samples were prepared as follows: The aqueous solutions resulting from the washing procedure were lyophilized, and then approximately 1 mL of CD_3_OD was added to 15–20 mg of the resulting powder. The mixture was sonicated in an ultrasound bath for 15 min at 40 °C, and then it was transferred into an NMR tube for the analysis. The same procedure was applied to an aqueous solution of washing soap, used as reference. The other reference samples were prepared as follows: SemaSORB^®^ UV20107 was dissolved in CD_3_OD; re-PVB_1 and re-PVB_3 were dissolved in CD_3_OD after evaporating the ethanol solvent used for their formulation.

### 2.7. Washing Test

The coated textiles were washed according to the standard test method UNI EN ISO 105-C10:2008. The test method was intended to reflect the effect of washing using soap (i.e., liquid Marseille soap) in domestic and commercial laundering procedures on the textiles. The samples were washed with water and soap (liquor ratio of 1:50 *wt/v*) at 40 °C for 30 min under stirring conditions; then, they were dried in the oven at 60 °C for 30 min. The weight loss was calculated according to Equation (1):(1)$$Weight\;loss\;\left(\%\right)=\frac{W_{i}-W_{f}}{W_{i}}\times 100$$
where W_i_ and W_f_ were the initial and the final weights of the samples after one washing treatment, respectively.

### 2.8. Mechanical Test

The maximum force and the elongation at the maximum force for the textiles were determined according to the standard test method UNI EN ISO 13934-1:2013 (strip method), which is a benchmark for textile producers. Tests were carried out using a SHIMADZU Universal Testing Machine. In the case of orthogonal woven textiles, the samples were cut both in weft and warp directions. The sample width was equal to 50 mm (without fringes), while the test length was equal to 200 mm. According to the test method, a pretension of 2–5 N was applied to the samples before starting the measurement, determined on the basis of the mass per unit area of the samples; afterwards, the elongation rate was set at 100 mm/min up to rupture.

For clothing companies, the reference to a standard test method is important to evaluate the mechanical resistance of textiles to permanent deformation under applied stresses and subsequent uses; fabrics with optimal mechanical properties are essential for the durability, quality, and aesthetic appearance of garments.

### 2.9. Thermal and Water Vapor Resistance Test

The thermal and water vapor resistance was determined—under steady-state conditions—according to the standard test method UNI EN ISO 11092:2014 (sweating guarded hotplate test). The samples were placed on an electrically heated plate with conditioned air ducted to flow across and parallel to its upper surface as specified in the standard test method. The thermal resistance (R_ct_) of the textile materials was determined according to Equation (2):(2)$$R_{ct}=\frac{\left(T_{m}-T_{a}\right)\cdot A}{{H-\Delta H}_{c}}-R_{ct0}$$
where T_m_ was the heating plate temperature (i.e., 35 ± 0.1 °C), T_a_ was the air temperature (i.e., 20 ± 0.1 °C), and A was the surface of the measuring plate (m^2^). In addition, H was the heating power supplied to the measuring plate (W), ΔH_c_ was the heating power correction in the case of measuring the thermal resistance (W), and R_ct0_ was the instrument constant for measuring the thermal resistance (i.e., 0.038 °C·m^2^/W). For this measurement, the relative humidity of the air was equal to 65 ± 3% and the airflow speed was equal to 1 ± 0.05 m/s.

For the determination of the water vapor resistance, a water-vapor-permeable but liquid-water-impermeable membrane covered the heated porous plate so that no liquid water contacted the sample. The water vapor resistance (Ret) of the textile materials was determined according to Equation (3):(3)$$R_{et}=\frac{\left(p_{m}-p_{a}\right)\cdot A}{{H-\Delta H}_{e}}-R_{et0}$$
where p_m_ was the saturated water vapor partial pressure (Pa) at the surface of the plate at temperature T_m_ (i.e., 35 ± 0.1 °C), p_a_ was the partial pressure (Pa) of the water vapor in the measuring chamber at temperature T_a_ (i.e., 35 ± 0.1 °C), and A was the surface of the measuring plate (m^2^). In addition, H was the heating power supplied to the measuring plate (W), ΔH_e_ was the heating power correction in the case of measuring the water vapor resistance (W), and R_et0_ was the instrument constant for measuring the water vapor resistance (i.e., 4.30 Pa·m^2^/W). For this measurement, the relative humidity of the air was 40 ± 3% and the airflow speed was equal to 1 ± 0.05 m/s.

### 2.10. Ultraviolet Protection Factor (UPF) Analysis

The UPF was determined according to the standard test method UNI EN 13758-1:2007. This European Standard specifies a method for the determination of the erythemal-weighted ultraviolet radiation transmittance of standard conditioned apparel fabrics to assess their solar UV protective properties. This method is not suitable for fabrics that offer protection at a distance such as umbrellas or shade structures. The UPF of a textile material is determined from the total spectral transmittance T(λ) via Equation (4):(4)$$UPF=\frac{\sum_{\lambda =290}^{\lambda =400}E\left(\lambda \right)\varepsilon \left(\lambda \right)\Delta \lambda }{\sum_{\lambda =290}^{\lambda =400}E\left(\lambda \right)T\left(\lambda \right)\varepsilon \left(\lambda \right)\Delta \lambda }$$
where E(λ) is the solar spectral irradiance (in W/m^2^nm) of the solar summer spectrum measured at Albuquerque, ε(λ) is the relative erythema action spectrum, Δλ is the fixed wavelength interval of the measurements, and T(λ) is the spectral transmittance as a function of the wavelength λ. E(λ) and ε(λ) are defined by the CIE International Commission on Illumination (CIE Research Note, 1987). The solar UV spectrum as measured at the earth’s surface extends between 290 nm and 400 nm.

The arithmetic mean of the UVA transmittance for the textile was calculated as follows (Equation (5)):(5)$$UVA=\frac{1}{m}\sum_{\lambda =315}^{400}T\left(\lambda \right)$$

The arithmetic mean of the UVB transmittance for the textile was calculated as follows (Equation (6)):(6)$$UVB=\frac{1}{k}\sum_{\lambda =290}^{315}T\left(\lambda \right)$$
where T(λ) was the spectral transmittance as a function of the wavelength λ, while m and k were the number of measurement points between 315 nm and 400 nm and between 290 nm and 315 nm, respectively.

The apparatus irradiated the sample with a parallel beam and collected all transmitted radiation with an integrating sphere. For uniform materials, at least four specimens per type of textile (treated and untreated) were prepared. The total spectral transmittance was measured by irradiating the samples with monochromatic or polychromatic UV radiation and collecting the total (diffuse and direct) transmitted radiation.

## 3. Results and Discussion

### 3.1. The re-PVB Formulations

To obtain an effective re-PVB coating on the fabric, the polymer was first dissolved into ethanol to be then applied on the textile substrate. However, to obtain a coating that was durable and resistant to washing, the re-PVB was chemically crosslinked by an isocyanate reagent. In addition, to give UV-protective properties to the final product, a UV absorber additive was tested in the re-PVB formulation. All these steps will be discussed in the following paragraphs, starting from the assessment of the compatibility of the formulations with textile processing, the investigation of the chemical reactions involved, and finally the effectiveness of the coating preparation.

As initial evidence, the solubility of the re-PVB in ethanol was confirmed [14] and all the prepared re-PVB formulations (i.e., re-PVB_1, re-PVB_2, and re-PVB_3) appeared homogeneous.

Considering the objective of processing the re-PVB formulations through spray coating, rheological analyses were carried out to explore their visco-elastic properties and their potential variation according to different stress and temperature conditions. During the spray coating process, the re-PVB formulations were applied to the fabrics at a temperature of around 40 °C.

According to the literature [34,35,36], the rheological properties of re-PVB formulations significantly influence the formation and size of the droplets resulting from the spraying process, thus potentially dictating the degree of homogeneity of the coating. It is known that low-viscosity values not exceeding tens of mPa·s are preferable for this type of process in order to ensure the proper atomization of the formulation and a more homogenous deposition.

The viscosity of the re-PVB formulation before (re-PVB_0) and after the addition of the crosslinker (re-PVB_1) and the UV absorber (re-PVB_3) were thus registered and compared to further explore the possible influence of the additives on the final rheological properties of the material. For the formulations containing the UV absorber, only the highest concentration was tested as it was considered the most representative of a potential change in rheological properties.

As represented in Figure 3, the analysis showed no significant differences in the rheological properties of the three re-PVB formulations subjected to different stress and temperatures. For all the tested systems, Newtonian behavior was observed and the average value of the viscosity was measured to be between 0.06 and 0.08 Pa·s, regardless of temperature variation.

These data prove that the introduction of the additives at the defined concentrations did not affect the viscosity of the re-PVB formulations, indicating the potential stability of the developed systems even during processing. The low values of viscosity registered confirm the suitability of the formulations with the spray-coating technique, while the preservation of the rheological properties indicates the absence of interactions between the re-PVB and both the crosslinker and UV absorber, avoiding any unexpected reactions during the fabrication process.

### 3.2. The re-PVB Crosslinked Films

The reaction of alcohols with isocyanates results in the formation of carbamate bonds. RUCOcoat FX 8041 containing blocked isocyanate functions could react with the hydroxyl group of the re-PVB after a heating step in an oven, fostering the formation of a three-dimensionally crosslinked film. According to the datasheet, the drying temperature was set at 160 °C to activate and promote the crosslinking reaction by de-blocking the isocyanate groups, which usually occurs at temperatures above 120 °C. In this way, a homogeneous, transparent, and colorless re-PVB_1 film was produced.

FT-IR analysis of the pristine re-PVB and the re-PVB_1-crosslinked film was carried out to determine their chemical compositions and structures. The FT-IR spectra of the re-PVB and re-PVB_1 film are shown in Figure 4, and the main absorption bands for the pristine re-PVB and re-PVB_1-crosslinked film are given in Table 1. The creation of a crosslinked polymeric film was confirmed by the re-PVB_1 film spectrum, in which the shape of the band at 1735 cm^−1^ changed after the crosslinking of the re-PVB with the isocyanate. It can be assumed that the band was given by the overlap of the stretching vibration of the C=O bond present in the re-PVB plasticizers and in the urethane groups, formed by the reaction of the unblocked isocyanate and the –OH moieties of the re-PVB. The content of vinyl acetate (VAc) units was too low in the re-PVB to observe its typical signals, such as a band at about 1700 cm^−1^ corresponding to the stretching vibration of the carbonyl group. In addition, the appearance of a new absorption band at 1530 cm^−1^, corresponding to the -NH bending vibration of the urethane bridge, also suggested that the crosslinking reaction occurred. From the overlapping of the spectra, it was possible to also observe the presence of typical re-PVB signals in the re-PVB_1 spectrum, such as the characteristic absorption bands of the vinyl butyral units between 1150 and 1000 cm^−1^, corresponding to different vibrational modes of the six-member ring. It demonstrates that the heating step, necessary for the crosslinking reaction, did not cause any damage to the polymer matrix. Other characteristic signals of the urethane group could not be precisely assigned due to their overlap with the signals of the starting material. In fact, the broad absorption band at 3600–3200 cm^−1^ could be attributed to the stretching vibration of –NH groups formed after the crosslinking reaction but also to the stretching vibration of any unreacted –OH groups of the re-PVB. However, the absence of the typical band at 2270 cm^−1^ of the stretching vibration of the -NCO group indicated that all the available isocyanate had reacted [37]. Further considerations of the RUCOcoat FX 8041 spectrum were not possible since the exact composition of the commercial product is unknown.

Similar considerations could be drawn also for the re-PVB_2-crosslinked film and the re-PVB_3-crosslinked film. In fact, to compare the spectrum of the two crosslinked films to the spectra of the re-PVB and the re-PVB_1-crosslinked film (Appendix A), not only the typical signals of re-PVB were present but also the signals discussed above as characteristic for the occurred crosslinking reaction between the unblocked isocyanate and the alcohol groups. This result proved that the addition of a small amount of the organic UV absorber molecule did not jeopardize the crosslinking reaction, successfully leading to the formation of the film. However, considering the complexity of the analyzed systems, it was not possible to detect or distinguish all the typical signals of the organic UV absorber. This could have been due to its low content (<1 wt.%) within the two formulations. Even if the re-PVB_2 and re-PVB_3 films appeared homogeneous and transparent like the re-PVB_1 film, they were slightly yellowish due to the presence of the organic UV absorber. This factor could potentially limit their use in textile processing depending on the final application (e.g., depending on the desired color of the ultimate clothes).

### 3.3. The re-PVB-Coated Fabrics

The spray coating was a rapid technique of application, which led to the creation of a homogeneous, transparent, and flexible re-PVB-coated fabric, potentially replicable in the production of technical textiles (Appendix A). Some of these features are clearly visible in Figure 5, where optical microscope images of the uncoated and re-PVB_1-coated fabrics are reported. Compared to other coating application techniques (e.g., dip coating), the re-PVB coating remained predominantly on one side of the fabric, reducing its stiffness. The re-PVB_1 formulation, as well as the re-PVB_2 and re-PVB_3 ones, were successfully applied to the fabrics through the spray technique. The presence of the coating onto the substrate was confirmed by FT-IR analysis of the treated textile samples (Figure 6). In fact, the FT-IR spectra showed the same peaks registered in the previous spectra of the re-PVB films (Figure 4 and Appendix A), described in detail in Table 1. Instead, the typical signals of the polyester fabric were no longer visible, such as the sharp peak at 1740 cm^−1^ due to the stretching vibration of the C=O bond and all the principal signals between 1250 and 730 cm^−1^ due to different types of vibrations of the polyester’s chemical groups [44,45]. This indicated that all the samples were effectively coated on the surface.

The durability and stability of the re-PVB coatings on the fabric were evaluated through a washing test. The results pointed out the effectiveness of the coating process by registering a negligible loss in weight for all the coated samples compared to the average amount of coating applied on the fabric (i.e., 20 ± 3 wt.%) (Figure 7a). Thus, the creation of a crosslinked network into the polymer matrix enabled us to guarantee satisfactory stability. These considerations were further confirmed by the FT-IR spectra registered for all the samples after the washing test. Figure 7b shows the FT-IR spectra of the re-PVB_1-, re-PVB_2-, and re-PVB_3-coated fabrics after the washing test, compared to the spectrum of the uncoated fabric. For all the samples, it was evident that the optimal overlapping between the FT-IR spectra was a sign of the presence of the crosslinked re-PVB coating on the surface of the fabric, even after the washing test.

The SEM image of the re-PVB_1-coated fabric in Figure 8b reveals the presence of a continuous and uniform polymer layer on the surface of the textile. Compared to the uncoated textile (average diameter of the fibers ≈ 12.8 ± 0.3 µm) (Figure 8a), where all the single fibers are individually distinguishable, it is evident that the fibers of the re-PVB_1-coated fabric were glued together (it was not possible to calculate the average fiber diameter), resulting in a surface almost totally made of crosslinked re-PVB. This result is in accordance with the description made above, i.e., the crosslinking of re-PVB could effectively promote the formation of a coating on the textile substrate. In addition, the SEM analysis was performed on the sample after the washing test to evaluate the coating structure (Figure 8c). It was evident that the continuous crosslinked re-PVB_1 coating was still present on the fabric, avoiding any type of negative modification (e.g., superficial defects). The same considerations could also be made for re-PVB_2- and re-PVB_3-coated samples (Appendix A).

The wash water was characterized, upon lyophilization, through ^1^H NMR analysis. Furthermore, the spectra of the re-PVB_3, washing soap, and SemaSORB^®^ UV20107 were recorded. The results for the two most interesting regions of the spectrum, i.e., the aromatic region and the low-field part of the aliphatic region, are shown in Figure 9. It is possible to observe that, as expected, the major component in all the wash water samples was the washing soap itself. However, the signals related to the UV absorber SemaSORB^®^ UV20107 were also found in the wash water of the fabrics coated with the re-PVB_2 and re-PVB_3, indicating partial leaching during the washing process. The same observation can be applied to the plasticizer present in the PVB formulation, which is here identified as tri(ethylene)glycol-bis(2-ethylexanoate) [12] (see Appendix A for the full-size spectra), and it was detected in the wash water of the fabrics coated with all three re-PVB formulations, whereas it was, of course, absent in the uncoated fabric. Regarding the signals related to the re-PVB itself, visible as broad signals in the region 4.5–3.5 ppm (see the full-size spectra available in the Appendix A) [12], they were not found in the wash water samples, meaning that the leaching was likely limited to a small fraction of the UV absorber and the plasticizer. These results further confirmed the effectiveness of the re-PVB coating application on the textile.

The mechanical performance of the uncoated and coated textiles was evaluated by tensile strip tests. Woven fabric is produced by the interlacement of two sets of yarn: one is called the warp yarn (longitudinal), and the other one is the weft yarn (transverse). Depending on how these threads are woven, the mechanical properties of the fabric may change and may be different considering the warp or the weft direction. In design, mechanical properties are important to evaluate the resistance of textiles to permanent deformation under applied stresses and subsequent uses. Then, optimal mechanical properties are essential for the durability, quality, and aesthetic appearance of the garments. The re-PVB coatings could affect the maximum force (F_max_) at the specimen failure and the relative elongation at the maximum force (ε_max_), as evident from Figure 10. Summarizing the results, the following maximum force percentage changes were observed compared to the uncoated fabric for the re-PVB_1-coated fabric (warp: −0.2%; weft: +15%), re-PVB_2-coated fabric (warp: +2%; weft: +21%), and re-PVB_3-coated fabric (warp: +3%; weft: +20%). While the maximum force in the warp direction was nearly the same for all the samples, the presence of a re-PVB superficial layer increased the F_max_ value in the weft direction with respect to the uncoated fabric. In addition, the following elongation at the maximum force percentage changes were observed compared to the uncoated fabric for the re-PVB_1-coated fabric (warp: +19%; weft: +18%), re-PVB_2-coated fabric (warp: +22%; weft: +23%), and re-PVB_3-coated fabric (warp: +21%; weft: +23%). To compare the re-PVB-coated fabrics to the untreated one, an evident increase in the elongation at the maximum force was observed in both weaving directions (i.e., warp and weft). As an example, Figure 11 shows the typical graphic trend for each sample. It is evident that the mechanical properties along the warp and the weft directions were different for each sample. It Is worth mentioning that the presence of re-PVB affected the mechanical behavior strongly even at a low deformation (Figure 11). Indeed, at a low deformation, the measured force for the coated fabrics was higher than for the pristine textile, evidencing a higher level of stiffness, which was clearly perceived qualitatively. In addition, focusing on the first part of the curve, a change in the slope is registered, due to the initial relative movement of the warp and the weft yarns. This phenomenon is encountered with textile fabrics, depending on the construction parameters of the fabric in question (e.g., weave, yarn density, cover factor) [46]. This effect was particularly evident in the uncoated fabric, whereas for the re-PVB-coated samples—where the re-PVB acted as a glue between the yarns—it was less visible. Overall, the presence of re-PVB coatings made it possible to improve the mechanical resistance of the textile material when subjected to applied loads. In fact, a finishing process is one of the factors that can affect the mechanical properties of woven fabrics, including yarn properties, the fabric weave, and the thread density. However, the addition of the organic UV absorber in the re-PVB coating formulation (i.e., the re-PVB_2 and re-PVB_3) did not lead to marked changes in the mechanical properties.

The thermal (R_ct_) and the water vapor (R_et_) resistance of the uncoated and coated fabrics were evaluated by the sweating guarded hotplate test. The thermo-physiological comfort of a textile material is an important aspect for both producers and customers, especially in the sportswear sector. In fact, a garment must ensure an optimal heat balance—thanks to the proper exchange of heat and moisture—between the human body and the external environment. This is a complex phenomenon that is affected by different factors, such as the characteristics of the fibers (e.g., composition and morphology), the type of fabric (e.g., structure and finishing treatments), and the nature of the final garment (e.g., fit and size). Therefore, it is not easy to know a priori the performance of a fabric or a garment, but the modulation of those factors makes It possible to develop a product that best reflects the desired properties [47,48]. The re-PVB coatings could affect the thermal and water vapor resistance of the pristine fabric, as evident from Figure 12. Compared to the uncoated fabric, the thermal resistance of the re-PVB-coated samples registered a reduction due to the presence of the re-PVB, which had a thermal conductivity of 0.236 W/mK [14]. As described before, the thermal resistance of a fabric could depend on different parameters, including the type of material (thermal conductivity of polyester = 0.127 W/mK) [49] and the structure and the thickness of the fabric. Thus, the application of a polymeric layer characterized by a higher thermal conductivity than the pristine material (i.e., polyester) led to a final reduction in the thermal resistance. On the contrary, the presence of a continuous superficial layer brought an increase in the water vapor resistance. The re-PVB coating could fill the voids between the fibers, as a result of which the fabric could absorb less water and transmit less water vapor through the fabric. The differences between the re-PVB-coated samples could be due to a slight variation in the weight of the coated samples and not to the addition of the organic UV absorber. The test results revealed a slight worsening of the thermal and water vapor resistance values of the coated fabrics compared to the pristine one, though not to a degree that would significantly impact the thermo-physiological comfort of the textile, suggesting that these enhanced UV-protective coatings could remain appropriate for use in commercial garments.

According to the standard test method UNI EN 1 3758-1:2007, the mean UPF of the coated samples was evaluated, as well as their average UVA and UVB transmittance. Figure 13 and Table 2 summarize the results for the re-PVB_1-, re-PVB_2-, and re-PVB_3-coated samples, compared to the uncoated fabric.

Even if the uncoated textile already had an optimum UPF according to the UNI EN 13758-1:2007 standard, the application of the coatings showed a great improvement of the UPF value, demonstrating the potential and the effectiveness of the developed coatings in UV protection. These coatings can also be successfully applied to other textiles that do not have enough UV radiation protection. This was a promising result for the selected synthetic textile, potentially replicable on other fabrics characterized by other compositions, other structures or other colors. The measured UPF values showed that even the re-PVB_1-coating was able to guarantee good UV protection. In the literature, it has been reported that PVB has shown a strong UV absorption band at 415 nm and a small shoulder at 390 nm [14]. Furthermore, although the absorption of PVB is theoretically low in the investigated range of UV light (<400 nm), it must be considered that different additives (<1 wt.%) are normally incorporated into commercial PVB for laminated glass, including UV absorbers and light stabilizers [12,14,16]. The residual presence of these substances in re-PVB could justify the good UV-protective property of the re-PVB_1-coated fabric. Nevertheless, the addition of the organic UV absorber—even in low quantities—allowed us to almost duplicate the UPF value of the coated fabric. An effective UV absorber must be able to absorb in the entire UV range, to remain stable under UV radiation, and to dissipate the absorbed energy without degradation [50]. According to the technical datasheet (see Appendix A), the organic UV absorber SemaSORB UV20107 exhibited strong absorption in the UV range below 360 nm (A_max_ at λ = 320 nm) and a large reduction in absorption in the visible range above 400 nm. The main disadvantage of the selected additive could be the high transmission in the UV range of 360 < λ < 400 nm, but the results reported in Table 2 show that the UVA transmittance remained extremely low, ensuring excellent UV protection. Effective UV-protective textile materials should guarantee a diffusive transmission below 5% over the whole UV range up to 400 nm [13]. As evident from Table 2, the percentage of transmitted UVA and UVB radiation remained extremely low, ensuring excellent protection over the entire UV range.

## 4. Conclusions

In this work, we have successfully implemented a circular approach to the repurposing of PVB as a functional coating on textiles, exploiting its outstanding properties as a film, after efficient separation and recovery from laminated glass, where it is mainly used. The addition of an organic UV absorber into the re-PVB formulation could enhance the UV protection performance of the final coated textile. Thus, the re-PVB formulations were applied through a two-step spraying treatment on fabric, successfully obtaining a crosslinked functional coating on the textile surface. The effectiveness of the treatment was confirmed by FT-IR spectroscopy, while SEM analysis revealed the presence of a continuous and uniform polymer layer on the surface of the fabric. The durability and stability of the re-PVB coatings were evaluated through a washing test, registering a negligible loss in weight for all the coated samples compared to the average amount of coating applied on the fabric (i.e., 20 ± 3 wt.%). The robustness of the re-PVB formulations towards washing was further confirmed by repeating the FT-IR and SEM analysis after the washing test, evidencing the absence of any negative modification. Complementary results were obtained by the analysis of the wash water through ^1^H NMR upon lyophilization. These results further confirmed the effectiveness of the re-PVB-coating application on the textile, limiting the leaching to a reduced fraction of the UV absorber and the plasticizer of the re-PVB. Overall, the presence of the re-PVB coatings made it possible to improve the tensile strength of the textile material when subjected to applied loads, as demonstrated by the increase in the maximum force (F_max_) in the weft direction and the elongation at the maximum force (ε_max_) in both weaving directions (i.e., warp and weft). Finally, the application of the re-PVB coatings showed a great improvement in the UPF value, demonstrating the potential and the effectiveness of the developed coatings in UV protection.

These promising results could open the way to the effective recycling and valorization of this largely employed polymer; it is not only for the selected synthetic textile but is potentially replicable on other fabrics of different compositions, structures, or colors. The developed coatings could find application in the technical textile sector, where factors such as excellent UV protection are required. In future research, the re-PVB coatings may also be differently functionalized for a plethora of purposes by adding a different UV absorber or other types of additives, like flame retardants, increasing the versatility of the material and the recycling process.

## Figures and Tables

**Figure 1 polymers-16-03439-f001:**
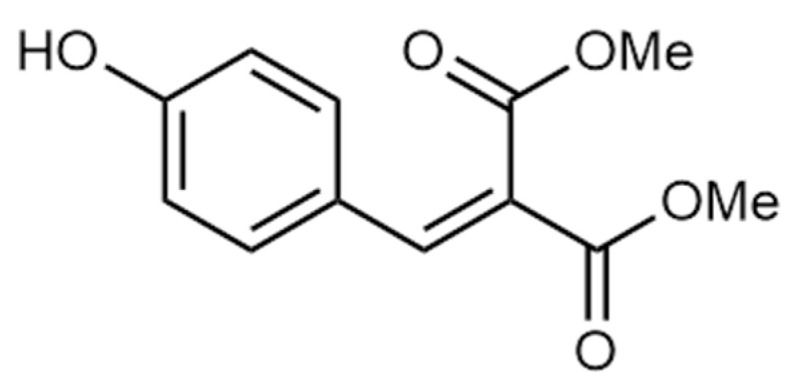
Chemical structure of UV absorber SemaSORB^®^ UV20107.

**Figure 2 polymers-16-03439-f002:**
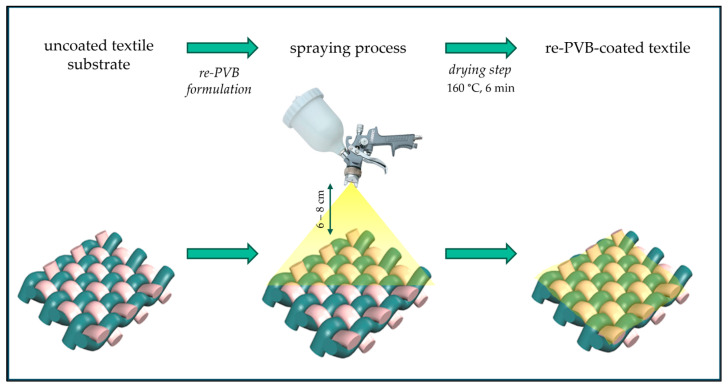
A scheme of the spraying process.

**Figure 3 polymers-16-03439-f003:**
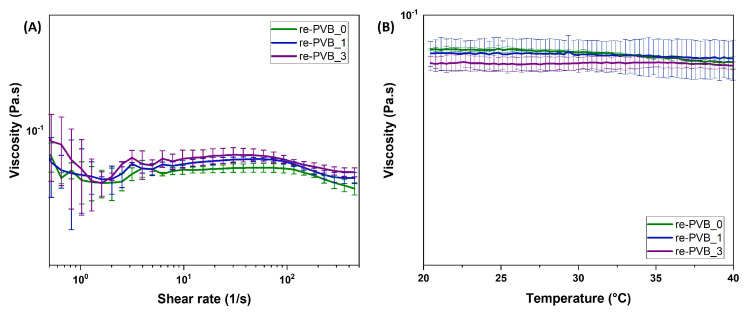
Viscosity of re-PVB formulations according to variation in shear rates (**A**) and temperature (**B**).

**Figure 4 polymers-16-03439-f004:**
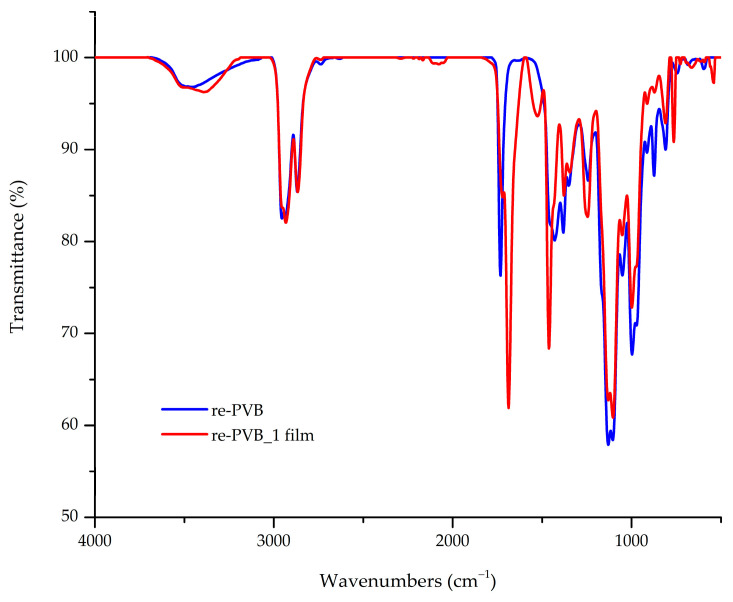
FT-IR spectra of the re-PVB_1-crosslinked film (red) and of the re-PVB (blue).

**Figure 5 polymers-16-03439-f005:**
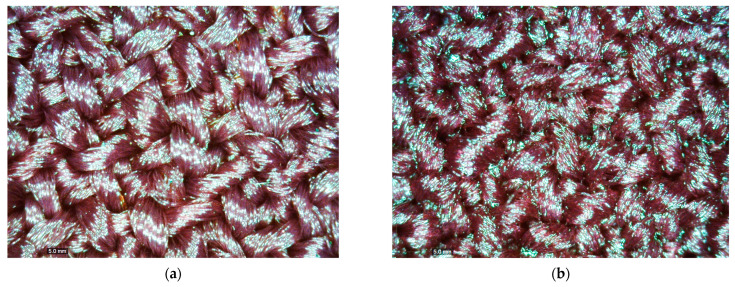
(**a**) Optical microscope image of the uncoated sample; (**b**) optical microscope image of the re-PVB_1-coated sample.

**Figure 6 polymers-16-03439-f006:**
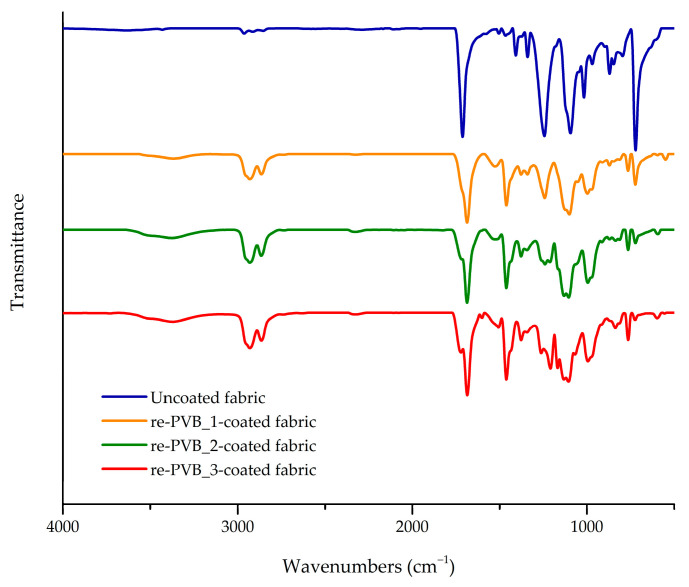
The FT-IR spectra of the re-PVB_1-(orange), re-PVB_2-(green), and re-PVB_3-(red) coated fabrics, compared to the spectra of the uncoated textile (blue). The spectra were recorded before the washing test.

**Figure 7 polymers-16-03439-f007:**
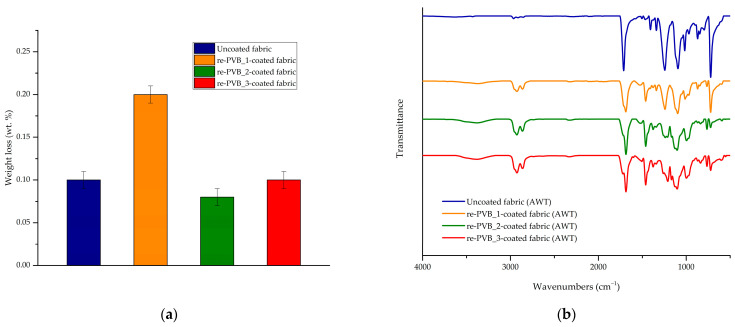
(**a**) The results of the washing test; (**b**) the FT-IR spectra of the re-PVB_1-(orange), re-PVB_2-(green), and re-PVB_3-(red) coated fabrics, compared to the spectrum of the uncoated fabric (blue). The spectra were recorded after the washing test (AWT).

**Figure 8 polymers-16-03439-f008:**
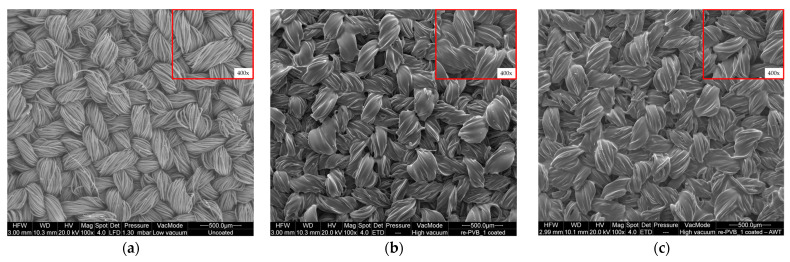
(**a**) An SEM image of the uncoated sample; (**b**) an SEM image of the re-PVB_1-coated sample before the washing test; (**c**) an SEM image of the re-PVB_1-coated sample after the washing test (AWT). All the images present a square section with a 400× magnification.

**Figure 9 polymers-16-03439-f009:**
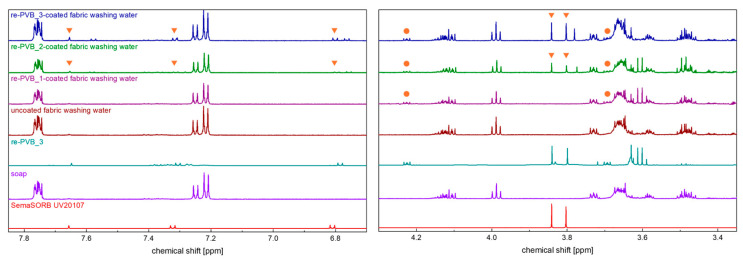
The ^1^H NMR spectra, in CD_3_OD solution, of the samples obtained by the lyophilization of the water samples used for the fabric washing. The spectra of the re-PVB_3, washing soap, and SemaSORB^®^ UV20107 are reported as references. In the spectra of the fabric samples, the signals related to the PVB plasticizer (●) and to the UV absorber (▼) are highlighted.

**Figure 10 polymers-16-03439-f010:**
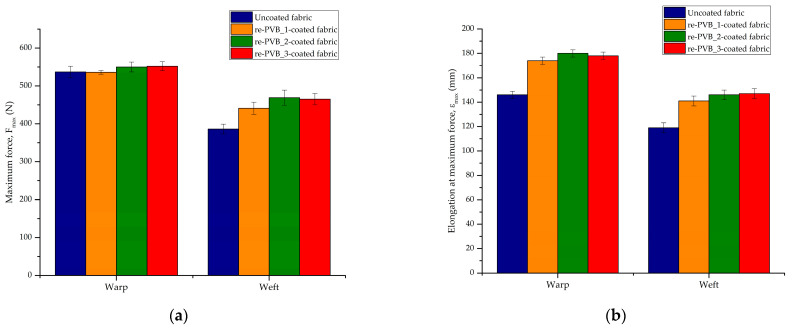
The results of the mechanical tests: (**a**) the maximum force F_max_ (N) of the uncoated and re-PVB-coated fabrics; (**b**) elongation at the maximum force ε_max_ (mm) of the uncoated and re-PVB-coated fabrics.

**Figure 11 polymers-16-03439-f011:**
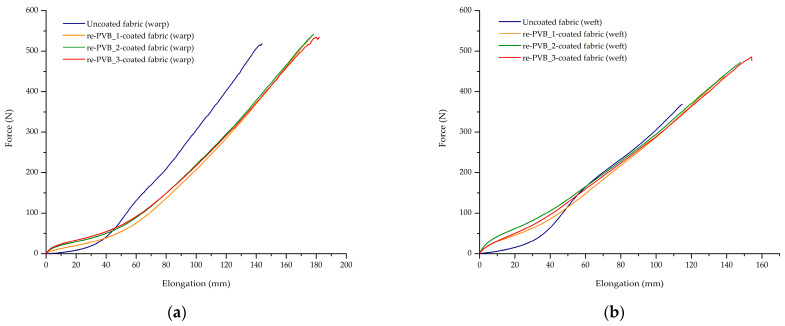
Force (N)–elongation (mm) graphs of uncoated and re-PVB-coated fabrics: (**a**) warp direction; (**b**) weft direction.

**Figure 12 polymers-16-03439-f012:**
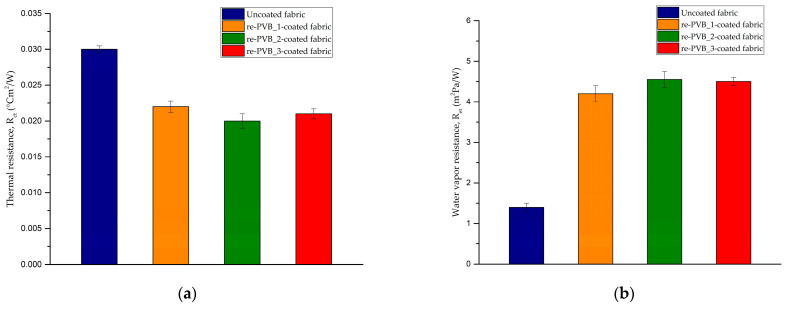
(**a**) The thermal resistance of the uncoated and the re-PVB-coated samples; (**b**) the water vapor resistance of the uncoated and the re-PVB-coated samples.

**Figure 13 polymers-16-03439-f013:**
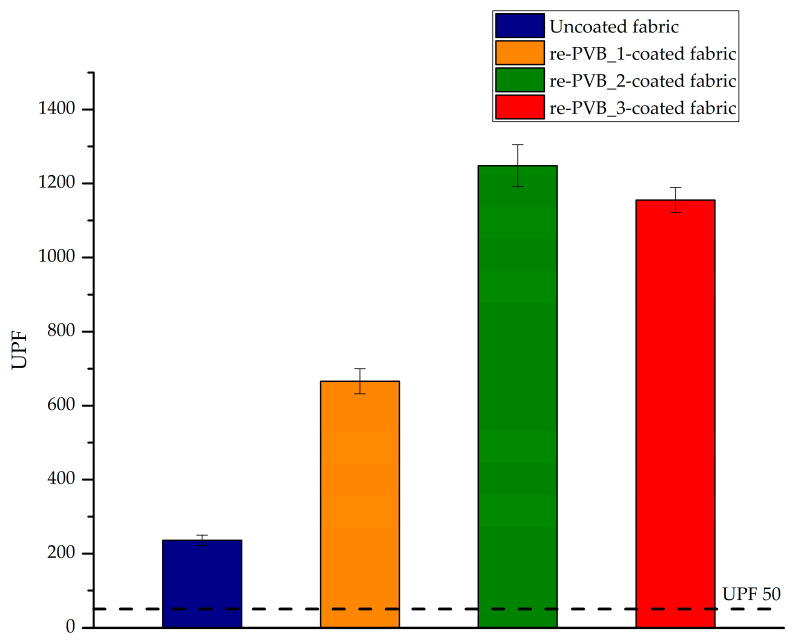
The UPF of the uncoated sample and of the re-PVB_1-, re-PVB_2-, and re-PVB_3-coated samples.

**Table 1 polymers-16-03439-t001:** FT-IR absorption bands associated with pristine re-PVB and re-PVB_1-crosslinked film.

Material	Wavenumber (cm^−1^)	Description	Ref.
re-PVB	3600–3200	O–H stretching (VA unit)	[12,14,15,38]
re-PVB_1 film	3600–3200	O–H and N–H stretching (VA unit + urethane group)	[12,14,15,38,39]
re-PVB	2960–2850	C–H stretching (alkyl groups)	[12,14,15,38]
re-PVB_1 film	2960–2850	C–H stretching (alkyl groups)	
re-PVB	1735	C=O stretching (plasticizers)	[12]
re-PVB_1 film	1735–1690	C=O stretching (plasticizers + urethane group)	[12,39,40]
Re-PVB_1 film	1530	CO–NH bending (urethane group)	[39,40,41,42]
re-PVB	1460–1350	CH_2_ and CH_3_ bending (alkyl groups)	[38,43]
re-PVB_1 film	1460–1350	CH_2_ and CH_3_ bending (alkyl groups)	
re-PVB	1140–1050	C–O–C stretching (VB unit + plasticizers)	[12,14,15,38]
re-PVB_1 film	1140–1050	C–O–C stretching (VB unit + plasticizers)	
re-PVB	1000	C–O stretching (VB unit)	[12,15,38]
re-PVB_1 film	1000	C–O stretching (VB unit)	

**Table 2 polymers-16-03439-t002:** UPF test results for re-PVB_1-, re-PVB_2-, and re-PVB_3-coated samples.

Sample	UPF	UVA Transmittance (%)	UVB Transmittance (%)
Uncoated fabric	236 ± 14	2.0 ± 0.1	0.20 ± 0.02
re-PVB_1-coated fabric	666 ± 34	0.90 ± 0.04	0.10 ± 0.01
re-PVB_2-coated fabric	1248 ± 57	0.5 ± 0.1	0.10 ± 0.01
re-PVB_3-coated fabric	1155 ± 34	0.60 ± 0.02	0.10 ± 0.01

## Data Availability

Data are contained within the article and Appendix A.

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
