# Peer review of "A UV-Protective Textile Coating Based on Recycled Poly(vinyl butyral) (PVB): A New Life for a Waste Polymer"

_polymers, 2024, doi:10.3390/polym16233439_

Round 1

Reviewer 1 Report

Comments and Suggestions for Authors

The current form of the manuscript is well formatted, provides valuable information, and the work is interesting. It may be accepted after some corrections.

  • The clarity of Figures 4 and 5 is not adequate, so consider providing bolder spectra for improved visibility.
  • Please provide bolded spectra and enhance the quality of images for all figures to make them more visually appealing.
  • Describe how FT-IR is used to confirm the presence of re-PVB and specify which peaks or functional groups are observed.
  • Discuss the SEM analysis of the coating’s surface morphology and explain its importance in assessing coating uniformity. Figure A and Figure C appear similar even after washing. The authors should clarify how much polymer remains after washing.
  • Outline the washing test protocol, including the number of cycles and washing conditions, to assess the durability of the coating.
  • Describe the mechanical tests performed, such as tensile or tear strength tests, and explain their relevance to textile applications.
  • Indicate whether any ISO or ASTM standards were followed for the coating process.
  • Provide details on the UPF testing method used and specify the range of UPF values achieved by the re-PVB-coated textiles.
  • Reduce similarity index of the manuscript, since it has very high 22%.

Comments on the Quality of English Language

Can be improved

Reviewer 2 Report

Comments and Suggestions for Authors

The authors investigated the potential of recycled PVB for the manufacture of textile coatings, especially with enhanced UV protection. The manuscript is well written and well-founded, with potential for publication in Polymers. Some details before publication:

>In the abstract, the authors were very general in presenting the main results. The main results must be reported, especially what enhances UV protection;

>There is currently a demand for less toxic materials for the manufacture of new products. Is the UV agent in Figure 1 from a clean source? Please indicate in the manuscript if this is the case;

>The authors in the FTIR discussion comment on the crosslinking process. Why didn't they perform the gel extraction test to assess the crosslinking content?

>Do the authors not have thermogravimetry analysis to assess the thermal stability of the compounds before and after coating? What temperature will the coating withstand to maintain its adhesive stability?

> “The spray coating is a rapid technique of application, which led to the creation of a homogeneous, transparent and flexible re-PVB coated fabric, potentially replicable in the production of technical textiles.” Would it be possible to add photos of the coated fabrics?

>Figure 7. The coating should help to make the fabric surface smoother. Did the authors perform any surface roughness tests to more robustly validate the SEM discussion? Did the authors not perform atomic force microscopy (AFM) testing with roughness data? Or contact angle to assess the difference with the coating?

> Did the authors not perform adhesion testing to assess the long-term stability of the coating?

Reviewer 3 Report

Comments and Suggestions for Authors

The submitted manuscript presents a UV-protective textile coating based on recycled poly(vinyl butyral). While the topic is relevant and timely, the study requires some fundamental revisions and additional experimental validation to meet the standards for publication in Polymers.

Below are key recommendations for improvement:

Justification for Fabric Choice: The authors are encouraged to provide a clear rationale for selecting a synthetic fabric with an already high UV protection factor (UPF 236) in its untreated state. Typically, UV-protective coatings are applied to textiles with lower UPF values to enhance their protective capabilities. Clarification on this choice is needed.

Wash Durability Testing: To accurately assess the wash resistance of the developed coating, a minimum of 10 washing cycles should be conducted, followed by a thorough evaluation of the fabric’s functional properties post-washing.

Impact on Air Permeability: The manuscript would benefit from additional analysis regarding the effect of the applied coating on the air permeability of the fabric, as this is a critical parameter affecting the comfort of the end product.

The abstract needs to be more informative, specifically by including the key results of the study. The current version lacks a summary of the main findings.

Round 2

Reviewer 2 Report

Comments and Suggestions for Authors

The authors satisfactorily addressed the questions raised in the first review. In addition, improvements and recommendations were included, improving the quality. The manuscript has merit for publication in Polymers.

Author Response

We thank the Reviewers for their evaluation and positive comments on our work, which was considered suitable for publication on Polymers after some revisions.